# Comparison of herpes simplex virus 1 genomic diversity between adult sexual transmission partners with genital infection

Molly M. Rathbun[1], Mackenzie M. Shipley[1], Christopher D. Bowen[1], Stacy Selke[2], Anna Wald[2,3,4,5], Christine Johnston[2,4,5], Moriah L. Szpara [1]*

1 Department of Biochemistry and Molecular Biology, Department of Biology, Center for Infectious Disease Dynamics, and the Huck Institutes of the Life Sciences, Pennsylvania State University, University Park, Pennsylvania, United States of America, 2 Department of Laboratory Medicine and Pathology, University of Washington, Seattle, United States of America, 3 Department of Epidemiology, University of Washington, Seattle, Washington, United States of America, 4 Department of Medicine, University of Washington, Seattle, Washington, United States of America, 5 Vaccine and Infectious Disease Division, Fred Hutchinson Cancer Research Center, Seattle, Washington, United States of America

* moriah@psu.edu

**Data Availability Statement:** All relevant data are within the manuscript and its Supporting Information files. Viral genomes are available at

## Abstract

Herpes simplex virus (HSV) causes chronic infection in the human host, characterized by self-limited episodes of mucosal shedding and lesional disease, with latent infection of neuronal ganglia. The epidemiology of genital herpes has undergone a significant transformation over the past two decades, with the emergence of HSV-1 as a leading cause of first-episode genital herpes in many countries. Though dsDNA viruses are not expected to mutate quickly, it is not yet known to what degree the HSV-1 viral population in a natural host adapts over time, or how often viral population variants are transmitted between hosts. This study provides a comparative genomics analysis for 33 temporally-sampled oral and genital HSV-1 genomes derived from five adult sexual transmission pairs. We found that transmission pairs harbored consensus-level viral genomes with near-complete conservation of nucleotide identity. Examination of within-host minor variants in the viral population revealed both shared and unique patterns of genetic diversity between partners, and between anatomical niches. Additionally, genetic drift was detected from spatiotemporally separated samples in as little as three days. These data expand our prior understanding of the complex interaction between HSV-1 genomics and population dynamics after transmission to new infected persons.

## Author summary

Herpes simplex virus 1 (HSV-1) causes chronic oral and genital infections for which there are currently no vaccines, and epidemiology trends indicate that an increasing number of primary infections occur in the genital anatomical niche. We provide a clinical genomics analysis of adult sexual transmission of HSV-1, in the context of new genital infections. We find that viral genomes are mostly conserved between transmission partners, while

GenBank, as Accessions ON007132-ON007164 (33 in total).

**Funding:** We acknowledge support from the Eberly College of Science and the Huck Institutes of the Life Sciences at Pennsylvania State University, as well as the National Institutes of Health (NIH) grant R01 AI132392 (MLS), grant R21 AI130676 (MLS, CJ) and grant P01 AI030731 (CJ, AW). This research was funded, in part, by a grant from the Pennsylvania Department of Health Commonwealth Universal Research Enhancement (CURE) program (MLS). MMR was supported by NIH T32 GM 102057-5 and the Pennsylvania State University Kern Graduate School Academic Computing Fellowship. The funders had no role in study design, data collection and analysis, decision to publish, or preparation of the manuscript.

**Competing interests:** I have read the journal's policy and the authors of this manuscript have the following competing interests: CJ reports funding from UpToDate (royalties), AbbVie (consulting), Gilead (consulting), MedPace (DSMB), NIH and CDC. AW reports funding from NIH, Sanofi (grants), Aicuris (consulting), X-vax (consulting), Auritec (consulting), GSK (grants), Merck (DSMB), Crozet (consulting), VIR (consulting), and UptoDate (royalties).

still exchanging within-host diversity between partners. We showed that this diversity can persist between distinct periods of viral reactivation, and involve multiple viral genotypes. This study significantly improves our understanding of the sources of HSV-1 genetic diversity, which will be critical in designing effective therapeutics against the wide range viral genotypes and disease severity.

## Introduction

Herpes simplex virus type 1 (HSV-1 or *Human alphaherpesvirus 1*; family *Herpesviridae*, sub-family *Alphaherpesviridae*) is a highly prevalent human pathogen that infects over 60% of the global population and causes a wide range of symptom severity [1]. Most frequently, people are either asymptomatic or present with mild, self-limited ulcerations in the oral or genital tract. In rare cases the virus is responsible for severe outcomes including keratitis and encephalitis. Several recent studies have employed deep sequencing to look for links between virus and host genomic signatures and the clinical presentation of HSV-1 infection [2–9]. Yet, the large, ~152kb dsDNA HSV-1 genome and genetic diversity of infections collected from around the globe (1–4% variation in nucleotide identity between viral genomes) has made a clinical genomics understanding of the virus elusive [10–13].

   HSV-1 infection is chronic and cycles between active replication at epithelial mucosa, and a non-replicative latent state within neuronal nuclei [14]. Periodically, the virus can reactivate from neurons and travel in a retrograde direction along neuronal processes to the mucosal surface. After lytic replication in epithelial cells, the virus is released, or shed, from the mucosal surface with or without symptoms or lesions. Shed virus can be detected via qPCR from skin, mucosal, or buccal swabs [15,16]. Each shedding event can be functionally defined as "an episode", which is bounded by days with no detectable viral shedding [16,17,18,19]. Additionally, infected persons can alternate between asymptomatic or symptomatic virus shedding at different temporal periods (e.g. within a shedding episode or across distinct shedding episodes) [16,18]. HSV-1 is transmitted during periods of symptomatic or asymptomatic shedding, upon contact with the mucosal surface from either the oral or genital anatomical niche. In several high-income countries HSV-1 has surpassed the distantly related viral species HSV-2 as the leading cause of new genital herpesvirus infections [1,20–22]. However, it is currently unknown if this epidemiological shift will have an effect on HSV-1 genetic diversity in either the oral or genital anatomical niche [1,20,21].

   Many studies have explored global HSV-1 genetic diversity, and studies continue to reveal new genotypes as more viruses are sampled world-wide [6,9,10,23,24]. Positive selection for adaptive traits in HSV-1 has been observed in cell culture, and in response to antiviral treatment [25–28]. However, it is unknown if such evolution occurs often enough during natural infections and/or transmission events to account for the observed variability between consensus-level HSV-1 genomes [10]. Each consensus genome represents the most common nucleotide sequenced at each position in the viral genome. The number of single nucleotide differences between randomly sampled and unrelated pairs of HSV-1 consensus genomes ranges from less than 50 to greater than 2,000 loci, within each ~152,000 bp genome [5–10]. In addition, each infection harbors a population of viruses with genomes that may harbor minor variants (MVs), or alternative alleles which differ from the consensus genome [5–9]. This within-host or within-infection diversity is also known as standing variation [11,12,29]. Population dynamics, such as population bottlenecks and expansions, can impact within-host genetic diversity by enabling rapid shifts in MV frequency [11,12,29]. The number of MVs

detected within individual infections with either oral or genital HSV-1 has varied from a range of less than five to greater than 300 MVs within a single viral sample [5–9]. However, the degree of contribution to within-host diversity from an individual's transmission partner, or other host-specific factors, is not yet known.

In two recent studies, we used deep sequencing of HSV-1 genomes sampled from familial transmission partners to explore the degree of conservation versus divergence in viral genomes after transmission to a new host [7,30]. Both cases indicated near-complete nucleotide conservation between each partner's consensus genomes, despite the estimated timing of each transmission event having occurred either decades before or only days before. In Shipley et al., 2019, we evaluated the within-host viral genetic diversity between a mother and her neonate, and found that the majority of MVs were shared, although a small number of MVs were unique to each partner. It is unknown if such conservation can also be expected from unrelated adult sexual partners, where the immune responses between individuals are more varied.

In this study, we conducted an HSV-1 comparative genomics analysis of five recent adult sexual transmission pairs. This included HSV-1 positive samples from both the "source" and "recipient" (newly infected) transmission partners, with the recipient partner presenting within 8 weeks of first-episode genital HSV-1 infection. We used target enrichment and Illumina deep sequencing to recover whole viral genomes directly from clinical swabs, without viral propagation in culture. Samples of genital and oral viral shedding included daily home-collected swabs at two- and eleven-months post infection, as well as clinic-collected lesion samples. We aimed to determine the impact of transmission to the observed consensus-level viral genotypes, and to the fluctuation of within-host MVs. By comparing partner infections over their first year of infection, we were able to assess the degree of conservation between sexual partners, while also detecting any host-specific adaptations. In a subset of partners, we were also able to examine viral genomes from both the oral and genital niches. Ultimately, our data indicated that HSV-1 transmission has more immediate impacts on within-host minor variants than on consensus genome diversity. These findings provide a context for future analyses that explore additional clinical factors such as viral shedding, duration of infection, and the host immune response.

## Results

### Each pair of participants provides a different example of adult sexual HSV-1 transmission

Recently-infected participants were enrolled for clinical study after presenting with first-episode genital HSV-1, or they were referred to the clinic by their sexual partners (Table 1). We defined first-episode infection by the absence of HSV-specific IgG antibody (see Methods for details). We investigated the HSV-1 genomic variability in these ten transmission partners using direct-from-participant (uncultured) sequencing of viral DNA collected via genital or oral swabs (Fig 1). Five of the ten partners (50%) identified as female (Table 1). Participants were a median of 23 years of age (range 19–38), with seven (70%) self-identified as white, two as Asian (20%) and one as mixed (10%) (Table 1). Participants enrolled a median of 29.5 days (range 12–70) after first-episode genital HSV-1 infection, and a median of 2,491 days (range 57–6,009) after their first self-reported episode of oral HSV-1 infection [31]. Nine out of ten participants presented with asymptomatic oral HSV-1 infection (all except participant 47, from Pair 4), though not all exhibited viral shedding during the study or shed sufficient amounts of viral DNA for sequencing. Genital shedding was detected in nine out of ten participants (all except participant 48, from Pair 5) (Table 1). In Pair 2, both partners enrolled with primary genital infections within one month of each other, and thus the self-reported directionality of transmission is uncertain (see Fig 1A).

**Table 1. Participant demographics and shedding rates.**

| Pair # Partner, Participant ID | Sex | Race / Ethnicity | Age | Infection status at screening | Genital Session 1, % positive | Oral Session 1, % positive[1] | Genital Session 2, % positive | Oral Session 2, % positive | Time since first genital symptoms | Time since first oral symptoms[2] |
|---|---|---|---|---|---|---|---|---|---|---|
| Pair 1 Source, 40 | M | Asian/Non-Hispanic | 19 | Non-primary | 21.4 | 25 | nd | nd[3] | Unknown | 14 years |
| Pair 1 Recipient, 41 | F | Asian/Non-Hispanic | 19 | UTD[4] | 14.8 | 7.4 | 0 | 3.4 | 57 days | 57 days |
| Pair 2 Source, 42 | F | Mixed/Non-Hispanic | 21 | UTD | 5.5 | 0 | 31.2 | 3.1 | 70 days | Unknown |
| Pair 2 Recipient, 43 | M | White/Non-Hispanic | 21 | Primary | 28.5 | 14.2 | 73.9 | 47.8 | 30 days | Unknown |
| Pair 3 Source, 44 | M | White/Non-Hispanic | 25 | Non-primary | 3.3 | 10 | nd | nd | 13 days | 7 years |
| Pair 3 Recipient, 45 | F | White/Unknown | 24 | Primary | nd | nd | nd | nd | 15 days | Unknown |
| Pair 4 Source, 46 | F | White/Non-Hispanic | 30 | Primary | 16.1 | 3.2 | 60.7 | 39.2 | 12 days | 16 years |
| Pair 4 Recipient, 47 | M | White/Non-Hispanic | 38 | Non-primary | 16.6 | 0 | 0 | 0 | 16 days | Unknown |
| Pair 5 Source, 48 | M | White/Non-Hispanic | 23 | Non-primary | 7.4 | 33.3 | nd | nd | Unknown | 76 days |
| Pair 5 Recipient, 49 | F | White/Non-Hispanic | 23 | Primary | 3.7 | 3.5 | 3.5 | 0 | 34 days | Unknown |

[1]Percent days positive for HSV genomes during 30-day daily self-swabbing sessions.

[2]Time since first symptoms to enrollment.

[3]nd, no data.

[4]UTD, unable to determine.

Swabs for viral DNA sequencing were either self-collected from the entire genital area or from a specific lesion site by a clinician (see Fig 1B and Methods for details). Swabs were collected via daily at-home collections during months 2 and 11 into the study, or directly in the clinic for symptomatic lesions (Fig 1A; see Methods for details). During the daily swab surveys, viral shedding varied between 0 to 73.9% of days being positive for genital shedding (mean 19.1%), and between 0 to 47.8% of days being positive for oral shedding (mean 12.7%) (Table 1). We sequenced samples collected at multiple time points and anatomical locations from each participant, to explore the potential for viral genetic variation to arise within distinct anatomical niches (i.e., oral vs. genital) and between different individuals (Fig 1B). Using oligonucleotide bait-based enrichment, Illumina deep sequencing, and a custom bioinformatics pipeline, we obtained full or partial consensus-level genomes for 33 out of 71 attempted participant samples (47% overall success rate) (Table 2). The success of viral DNA enrichment and genome sequencing was highly correlated with the quantity of starting HSV-1 DNA, which averaged over 1 million genome copies per sample (6.5 $\log_{10}$ copies per mL) in the 33 successful samples shown in Table 2 (see Methods for more details). The average sequencing coverage depth was ~3,000X across the viral genome, with 28 samples having average coverage depth $\geq$ 100X, and just 5 samples with $<$ 100X average coverage depth (Table 2).

## The consensus-level HSV-1 genome is unique to each transmission pair and is highly conserved between sexual partners

The average percent identity between world-wide, randomly sampled HSV-1 genomes has been estimated at ~97% [10]. We calculated the percent identity of viral genomes detected in

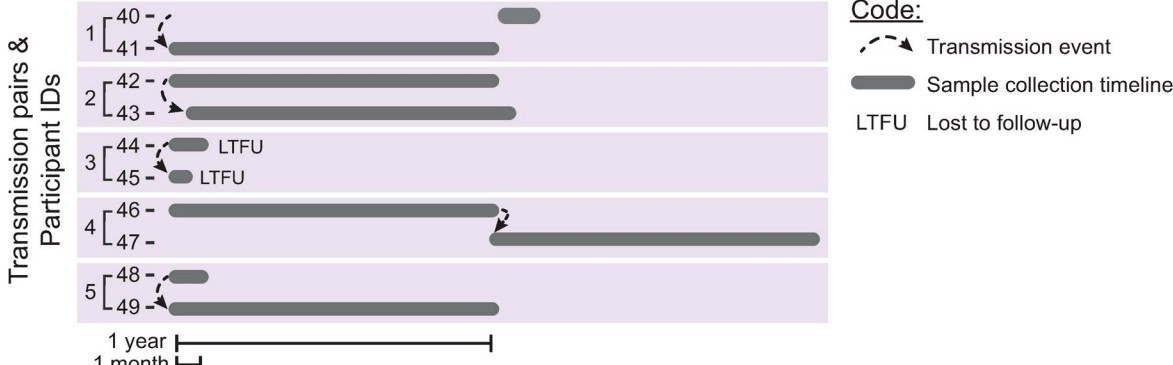

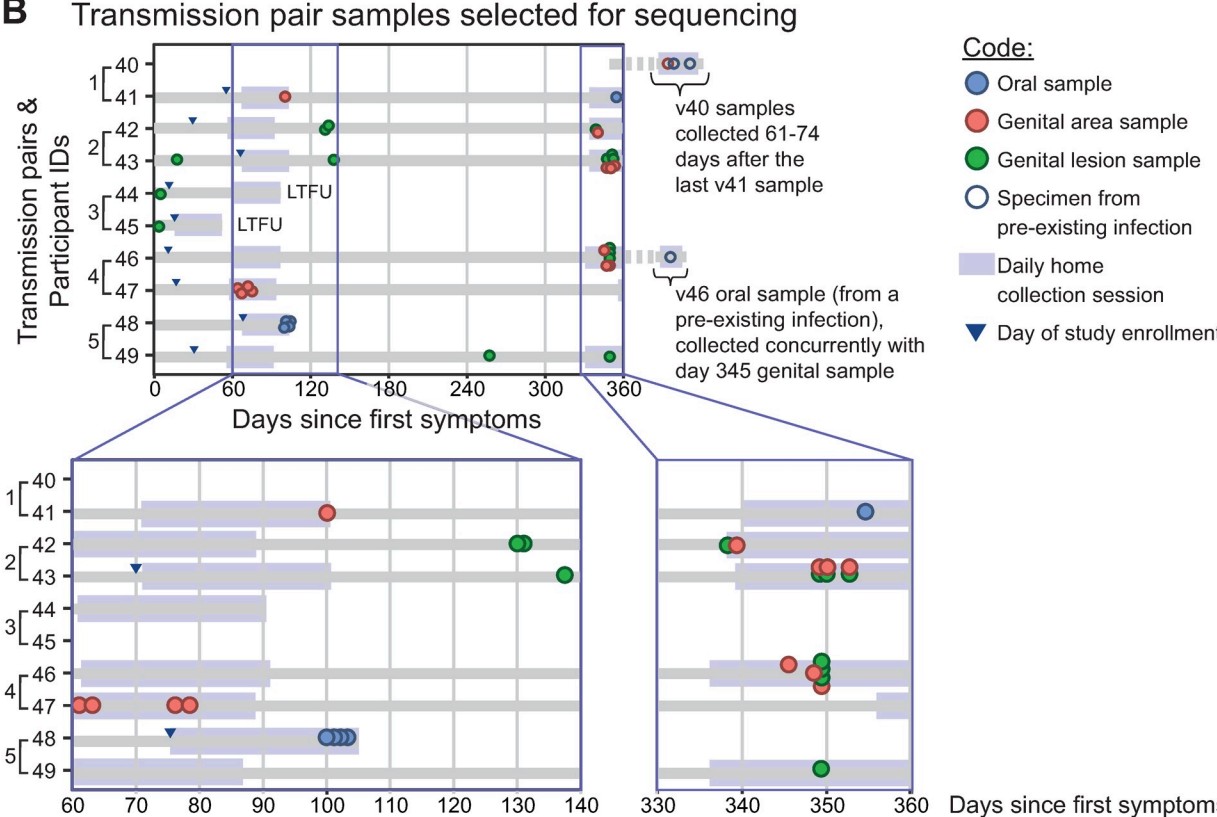

**Fig 1. Overview of samples sequenced from each transmission pair during a one-year period.** (**A**) Participants were enrolled for clinical study after the detection of first-episode genital infections. Transmission partners were referred to the study through their partners. In panel (**A**) the position of each grey bar indicates the relative calendar time frame of sampling for each participant, and the bar length reflects how long they were enrolled in the study. E.g., Participant 40 was referred as a source partner and thus completed only one month of daily swabs, whereas participant 41 enrolled with first-episode genital infection and completed a full year of the study. LTFU indicates participants lost to follow-up. (**B**) Sequenced samples are plotted by participant, according to the number of days since each participant's first reported symptoms. Dot color denotes sample type: oral area (blue), genital area (red), or site-specific genital lesion (green). Larger shaded bars early and late in each participant's first year of infection denote the 30-day sessions of daily self-collected swabs. Lesion samples were also collected at clinic visits at intervening times. Magnified panels in (**B**) highlight selected time frames, including the two sessions of daily self-collected swabs. Dots stacked vertically in these panels indicate samples collected on the same day. Open circles designate samples from pre-existing multi-year infections (v40 and v46 oral). Participant 40 (Pair 1, source) presented with a pre-existing oral infection, and a genital infection with an unknown start date. Participant 46 (Pair 4, source) entered the study with a pre-existing oral infection.

**Table 2. Thirty-three viral genome samples sequenced from five transmission pairs with adult HSV-1 infection.**

| Pair, Virus Genome ID | Sample ID | Time since initial symptoms | Sample type | HSV genomes in sample ($\log_{10}$ copies per mL) | Average coverage | Reads used for genome assembly[1] | Minor variants (MV) detected[2] | GenBank Accession |
|---|---|---|---|---|---|---|---|---|
| 1, v40 | v40_unk_gen | Unknown | Genital | 5.6 | 8,224 | 4.3M | 0 | ON007132 |
| | v40_y14_oral1 | 14 years | Oral | 4.5 | 428 | 220k | 27 | ON007153 |
| | v40_y14_oral2 | 14 years + 13 days | Oral | 6.8 | 7,315 | 3.9M | 0 | ON007154 |
| 1, v41 | v41_d100_gen | 100 days | Genital | 3.4 | 73^ | 34k | 0^ | ON007139 |
| | v41_d354_oral[D] | 354 days | Oral | 2.2 | 28^ | 4.1k | 530^ | ON007157 |
| 2, v42 | v42_d130_gen_les | 130 days | Genital lesion | 8.4 | 2,081 | 1.1M | 3 | ON007160 |
| | v42_d131_gen_les | 131 days | Genital lesion | 8.3 | 7,775 | 3.6M | 1 | ON007137 |
| | v42_d338_gen_les | 338 days | Genital lesion | 8.1 | 7,895 | 3.3M | 2 | ON007138 |
| | v42_d339_gen | 339 days | Genital | 6.2 | 274 | 131k | 2 | ON007164 |
| 2, v43 | v43_d17_gen_les | 17 days | Genital lesion | 6.1 | 191 | 87k | 2 | ON007141 |
| | v43_d137_gen_les | 137 days | Genital lesion | 6.9 | 1,648 | 749k | 1 | ON007156 |
| | v43_d349_gen | 349 days | Genital | 5.2 | 481 | 221k | 5 | ON007148 |
| | v43_d349_gen_les | 349 days | Genital lesion | 5.9 | 2,802 | 1.3M | 3 | ON007147 |
| | v43_d350_gen | 350 days | Genital | 7.2 | 2,476 | 1.1M | 1 | ON007152 |
| | v43_d350_gen_les | 350 days | Genital lesion | 8.6 | 1,118 | 517k | 1 | ON007159 |
| | v43_d352_gen | 352 days | Genital | 6.4 | 1,440 | 649k | 1 | ON007135 |
| | v43_d352_gen_les | 352 days | Genital lesion | 6 | 1,423 | 672k | 5 | ON007162 |
| 3, v44 | v44_d2_gen_les | 2 days | Genital lesion | 7.1 | 3,258 | 1.5M | 1 | ON007134 |
| 3, v45 | v45_d4_gen_les | 4 days | Genital lesion | 6.7 | 7,671 | 3.2M | 0 | ON007145 |
| 4, v46 | v46_d345_gen | 345 days | Genital | 6.9 | 820 | 349k | 0 | ON007144 |
| | v46_y16_oral[3] | 16 years | Oral | 5.6 | 34^ | 16k | 5^ | ON007133 |
| | v46_d348_gen | 348 days | Genital | 7.8 | 4,520 | 2.2M | 1 | ON007150 |
| | v46_d349_gen | 349 days | Genital | 8.3 | 5,062 | 2.4M | 1 | ON007146 |
| | v46_d349_gen_les1 | 349 days | Genital lesion | 7.7 | 3,797 | 1.6M | 0 | ON007158 |
| | v46_d349_gen_les2 | 349 days | Genital lesion | 7 | 9,222 | 4.0M | 1 | ON007143 |
| | v46_d349_gen_les3 | 349 days | Genital lesion | 8.4 | 7,333 | 3.2M | 0 | ON007151 |
| 4, v47 | v47_d61-79_gen[3,4] | 61, 63, 76, 79 days[4] | Genital | 2.6 | 31^ | 15k | 256^ | ON007163 |
| 5, v48 | v48_d100_oral[3] | 100 days | Oral | 6 | 350 | 158k | 62 | ON007155 |
| | v48_d101_oral[3] | 101 days | Oral | 6.1 | 466 | 216k | 234 | ON007136 |
| | v48_d102_oral[3] | 102 days | Oral | 6.2 | 198 | 136k | 0 | ON007161 |
| | v48_d103_oral[3] | 103 days | Oral | 6.4 | 78^ | 71k | 0^ | ON007140 |

*(Continued)*

**Table 2.** (Continued)

| Pair, Virus Genome ID | Sample ID | Time since initial symptoms | Sample type | HSV genomes in sample (log$_{10}$ copies per mL) | Average coverage | Reads used for genome assembly[1] | Minor variants (MV) detected[2] | GenBank Accession |
|---|---|---|---|---|---|---|---|---|
| 5, v49 | v49_d257_gen_les | 257 days | Genital lesion | 8.6 | 5,076 | 2.2M | 0 | ON007142 |
| | v49_d349_gen_les | 349 days | Genital lesion | 7.8 | 4,569 | 4.4M | 0 | ON007149 |

[1]When counting reads used for assembly, M denotes millions of reads and k denotes thousands of reads.

[2]The minor variant count listed here excludes those adjacent to homopolymers and repetitive elements (see Methods for details).

[3]Data files were pooled from two replicate sequencing runs.

[4]Four samples pooled before oligo-enrichment.

^Minor variant analysis performed with a stringent detection threshold of 20%, due to coverage depth <100X.

this study cohort by selecting the first genome sampled from each participant for comparison (Table 2). This resulted in an average nucleotide identity of 98.1% between the viral genomes from all ten individuals. In contrast, the percent identity between all viral genomes within each transmission pair (Table 2) was greater than 99% (n = 2 to n = 12 genomes per pair) for all pairs, except Pair 1 (98.3%, n = 5 genomes). Network graph analysis of all 33 viral genomes revealed that these samples spanned the previously observed genetic distribution of historical HSV-1 genomes–the vast majority of which have been isolated from oral infections (Fig 2; see S1 Table for list of comparison genomes) [5–7,10,30,24]. Samples from oral and genital niches within one participant, and from source and recipient partners within each pair, clustered together in this overall network graph analysis. Overall, the comparison of percent identity among viral genomes indicates a higher level of nucleotide conservation within each transmission pair than among randomly sampled infections world wide.

The within-pair analysis of percent identity indicated that 0.4–1.7% of the HSV-1 genomes between each of the five transmission pairs harbored nucleotide differences at the consensus-level (either as single nucleotide variants [SNVs] or at insertion and deletion sites [i.e. in/dels]). This included one or more consensus-level SNVs (outside of those at repetitive elements or in/dels) between the viral genomes in three out of five transmission pairs (Pairs 1, 2, and 4). Since these differences are not easily visualized in the larger network graph (Fig 2), we pursued a more detailed analysis by constructing a neighbor-joining phylogram (1,000 bootstraps) using a whole-genome alignment of the 33 genomes from transmission pairs in this study (Fig 3). Clusters of branches within pairs, such as Pair 2, indicated transient viral genome differences between partners, as well as over time and between genital vs. oral niches within a single participant's infection. The phylogram also highlighted transmission Pairs 1 and 4 as having the highest number of SNVs between at least one comparison of genomes amongst their respective partners. Analysis of the consensus genomes in Pair 1 indicated 268 SNVs between the genome from sample v41_d354_oral and all others from either partner. Likewise, within Pair 4 there were 51 SNVs detected between the genome from sample v47_d61-79_gen and all v46 genomes, and 8 SNVs between the genome from sample v46_y16_oral and all other genomes in this pair. Although the divergent genomes in these cases are the lowest-coverage samples within each pair, the SNV counts reported here exclude differences occurring at in/dels and repetitive elements, making these conservative estimates of genetic divergence for these samples. These single nucleotide differences demonstrate the viral genomic variability within and between transmission partners and oral vs. genital niches, which may be obscured when comparing these paired samples with the entire HSV-1 phylogeny.

## Network graph of globally sampled HSV-1 genomes and five new transmission pairs

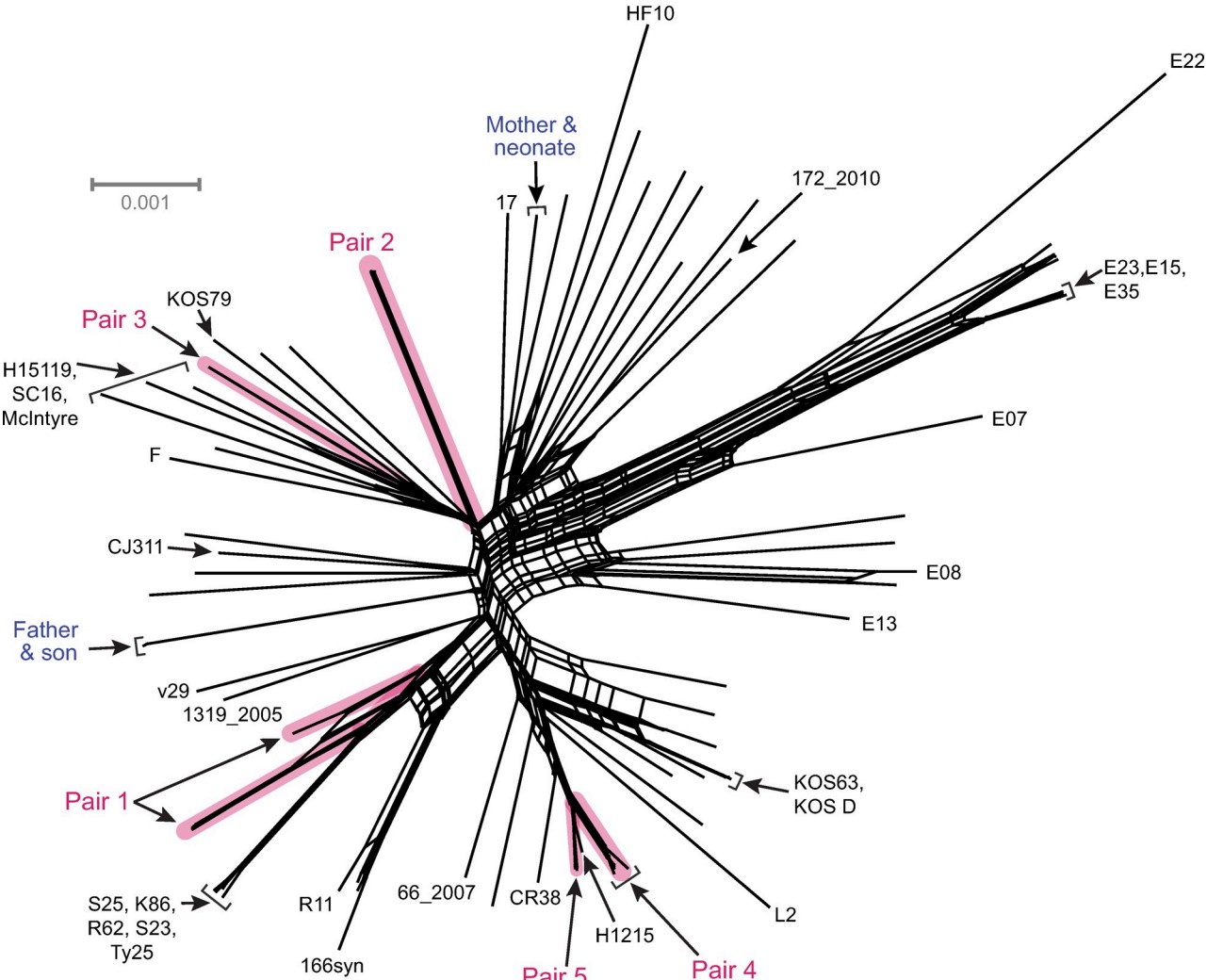

**Fig 2. Network graph comparing 65 globally sampled HSV-1 genomes and 33 viral genomes from new adult sexual transmission pairs (n = 10 participants).** The starburst pattern in this network graph (which excluded gaps) exemplifies the unique genetic diversity in each randomly sampled HSV-1 genome from around the globe (black branches). Consensus genomes sampled over time from individual participants, or between transmission partners, indicated near complete nucleotide conservation ($\geq$ 98% identity) and formed thick clusters of branches. Genome names in black denote well-known strains or isolates that are nearest-neighbors to the new transmission pair samples (highlighted in magenta). Genome names highlighted in blue indicate previously published data on viral genomes from parent-child familial transmission pairs. The gray scale bar indicates approximately 0.1% nucleotide divergence. Internal reticulations within the network reflect likely historical recombination events. All strain names and prior references for these are provided in S1 Table.

Transmission pairs with highly conserved viral genomes, such as Pair 3 and Pair 5, also provide an opportunity to study rare variants that are maintained through transmission. In the genital samples of Pair 3, the viral genomes were nearly identical between participants 44 and 45 (99.6% similar nucleotide identity), with the exception of indels at repetitive sites in the genome. Their shared nucleotide identity included a homopolymer frameshift mutation of $T_7$ (reference strain) to $T_8$ within the gene encoding glycoprotein H (gH, UL22), which removes a stop codon (Fig 4). This $T_7$ to $T_8$ frameshift changes the C-terminal amino acid tail from WRRE* to LETRIK, and if the next possible stop codon is considered, this mutation would

## Phylogram of HSV-1 genomes sampled from adult sexual transmission partners

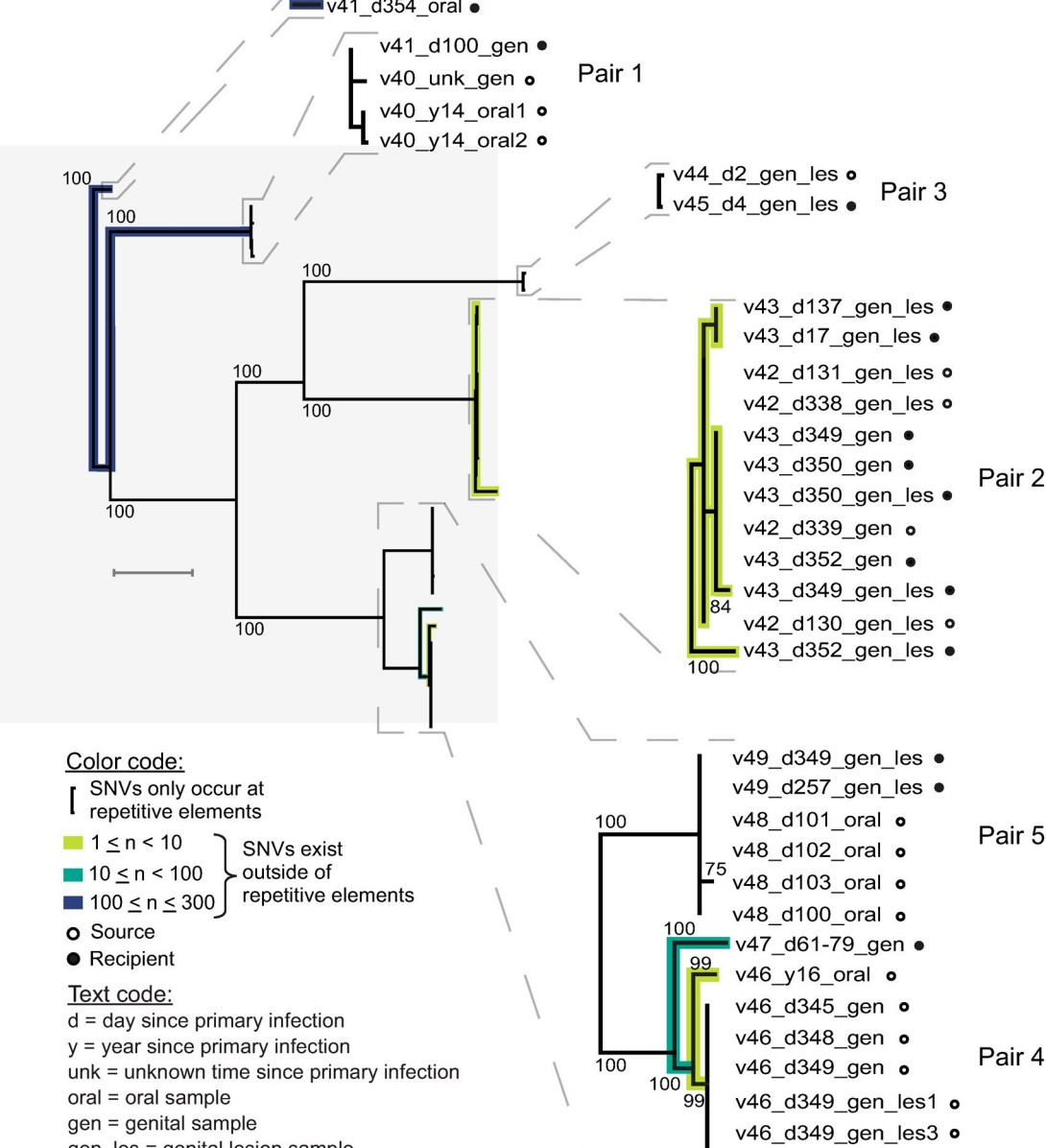

**Fig 3. Neighbor-joining phylogram reveals consensus-level differences among HSV-1 genomes from five transmission pairs (n = 10 individuals).** Single nucleotide differences or variants (SNVs) in the consensus-level viral genomes occur in three out of five transmission pairs (Pairs 1, 2, and 4; excluding those that occurred at repetitive elements). These SNVs demonstrate that genetic diversity may arise transiently between transmission partners and/or within individual infections. Branches in the phylogram are shaded according to the range of SNVs detected between genomes. Comparisons with a small number of SNVs ($1 \leq n \leq 10$, light green) are consistent with the genome conservation observed in cases of familial transmission. Comparisons with a larger number of SNVs ($100 \leq n \leq 300$, dark blue) are more similar to the level of divergence between globally sampled genomes in the larger network graph analysis (Fig 2). The gray scale bar indicates approximately 0.1% nucleotide divergence.

add an additional 14 amino acids (Fig 4). Only one other HSV-1 genome reported in the literature has shown the same mutation, which is the strain E07, sampled between 1981–1984 from a person in Nairobi, Kenya [10,32]. This extended open reading frame fits within the length of

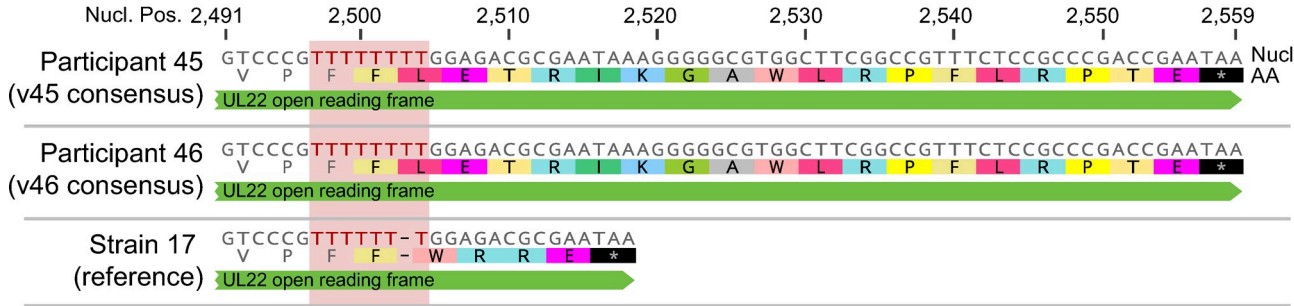

**Fig 4. A homopolymer frameshift mutation in viral gene UL22, encoding glycoprotein H (gH), occurs in both partners of Pair 3.** The consensus viral genomes from participants 44 and 45 (Pair 3) are identical, as there were no consensus-level single-nucleotide differences between these partners, outside of repetitive elements in the genome. The consensus sequences for the UL22 gene encoding gH of viral genomes v44 and v45 were compared to the HSV-1 strain 17 reference genome. This revealed a $T_7$ to $T_8$ homopolymer frame-shift mutation affecting the C-terminus of UL22. The T-homopolymer is indicated with red lettering and a pink background shading. This frameshift mutation ablates the canonical stop codon for gH, alters the four amino acid C-terminal tail found in all other HSV-1 isolates, and extends the encoded protein by 14 amino acids. Thus, the reference strain encodes a viral gH protein of 838 amino acids in length, while the viral genomes of both Pair 3 partners encode a predicted length of 852 amino acids for gH. The nucleotide position is numbered based on the position of this frameshift within the UL22 gene.

previously documented gH (UL22) transcripts, and it is followed by an additional poly-adenylation signal [33]. While the potential functional effects of this mutation are still unknown, it is noteworthy that this rare mutation occurs in one of the four essential glycoproteins for HSV-1 entry. The preservation of this variant C-terminal tail of gH between three unrelated individual hosts (i.e. the E07 isolate and both members of transmission Pair 3) suggests that this viral variant allele is functional for entry and spread. Length changes at similar homopolymer tracts, as well as large repeats, can expand the range of viral genetic diversity beyond just single nucleotide differences.

## The level of within-host HSV-1 diversity varies between participants and across sampling time

Several previous studies applying deep-sequencing of both cultured and uncultured HSV-1 samples have detected varying levels of within-host diversity, or minor variants (MVs), within a single sample from an infected host. These MVs indicate within-host variation, or the potential presence of multiple virus genotypes [5–9,30]. We analyzed each participant's viral population for the presence of MVs and compared them between transmission partners (Table 2). Overall, within-host diversity was detected in eight out of ten participants, with a wide range of MV count between samples (Table 2; see S2 Table for a detailed list of MVs in each sample). The average number of minor variants at non-repetitive sites across all samples with MVs detected was 19 (2% minimum threshold; excluding samples with zero MVs), which is consistent with other recent reports from uncultured clinical samples [5,7–9,23]. For those participants with temporal sampling, the data revealed fluctuations in the number of MVs detected within and between different shedding episodes (Tables 2 and S2).

A greater level of within-host diversity and MVs was observed in viral genomes isolated from Pair 1 (v40 and v41), Pair 4 (v46 and v47), and Pair 5 (v48) (Table 2). Pairs 1 and 4 were also the most divergent at the consensus-level between any transmission pair samples (Fig 3). In each participant, the positions of consensus-level single-nucleotide differences (i.e., SNVs) often coincided with positions of minor variants, suggesting that sites of within-host diversity

and MVs can influence the observed consensus-level genotype over time. We sequenced HSV-1 genomes from both oral and genital samples in at least one partner from each of these transmission pairs, allowing for analysis and comparison of virus populations in these distinct anatomical niches. Because five of the samples showing within-host diversity had less than 100X average coverage depth (i.e., below the requirement for our 2% MV detection threshold), we applied a more stringent requirement of $\geq$ 20% MV frequency for these lower-coverage samples (with a minimum coverage depth of 10X; Table 2).

## HSV-1 genetic diversity fluctuates between shedding episodes, both within and between transmission partners

We next explored the presence of coincident MVs in transmission partners, indicating potential transfer of viral populations during initial infection. As noted above, both partners in Pair 2 (v42 and v43) enrolled with primary genital infection, and thus the directionality of their transmission cannot be known with certainty. The transmission event that resulted in Pair 2 occurred within one month of study enrollment for both participants (Fig 1A), which allowed for an opportunity to examine genetic variation within the viral population of each individual almost immediately after infection was established. Within-host variation analysis of Pair 2 indicated less than 10 MVs in any individual sample from the inferred source, participant 42, and the recipient, participant 43 (Table 2 and Fig 5A). A maximum of three variants were detected in the v42 samples. These included one site in the gene UL27 encoding glycoprotein B (gB), where both G and A alleles were observed in fluctuating ratios from 80% to 40% G over two different shedding episodes (Fig 5B). The same genome position also showed within-host variation in the other partner in this pair, v43, in samples spanning three different shedding episodes (Fig 5B). During this time, the consensus-level viral genome for the v43 samples switched from having "G" as the dominant allele to having the "A" variant at 100% penetrance in the final two samples from this individual (Fig 5B). Both nucleotides (A and G) encode synonymous codons for thymidine at amino acid (AA) position 877 in gB. While relatively few MVs were detected in Pair 2 overall (Table 2), this single nucleotide site provides evidence for transmission of within-host HSV-1 genetic variation, with subsequent within-host changes in frequency across multiple shedding episodes.

Next, we compared viral genetic diversity present within overall genital-area samples versus site-specific genital lesion samples in the same shedding episode, to better understand the localization and/or spread of viral variants within an anatomical niche. A second minor variant elsewhere in the UL27 gene encoding gB was detected within v43 samples. In this case the nucleotide difference encoded a missense mutation in gB (Phe175Val). This minor variant reached a frequency sufficient to create a consensus-level difference (a SNV) between same-day swabs of the whole genital area, vs. site-specific genital lesion samples (Fig 5C). In the site-specific lesion sample v43_d349_gen_les, this minor variant encodes valine in a majority of the sequences (65%), while the same day genital area swab reflects only a 4% minor variant frequency of this allele. Two days later, another genital lesion sample (v43_d352_gen_les) revealed only 7% of viral genomes encoding valine, with the rest reflecting the phenylalanine seen in the same-day genital area swab (Fig 5C). In all other genital samples from this shedding episode, this locus encoded 100% phenylalanine. This change in minor variant frequency within a single host exemplifies how MVs can vary spatially within an anatomical niche, and over the temporal course of a single shedding episode.

Samples from Pair 2 also illustrate other potential sites of shared within-host diversity. For instance, the v43 samples from day 349 also harbor a minor variant in the UL2 gene, which encodes uracil-DNA glycosylase. In the genital area sample v43_d349_gen, there is a minor

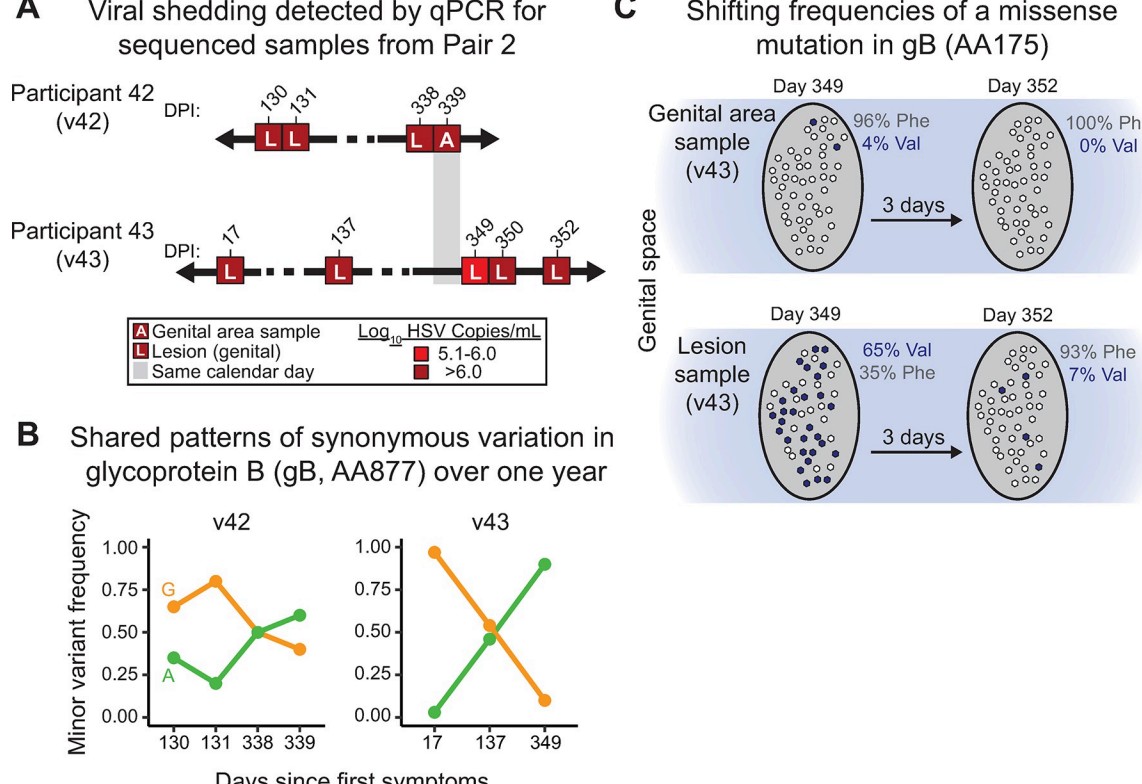

**Fig 5. With-host variation occurs in viral genome populations transmitted within and between partners over time.** (A) HSV-1 genomes were sequenced from Pair 2 specimens spanning multiple shedding episodes over the first year of infection. Analysis of within-host diversity in these samples revealed two MVs of interest within the gene encoding glycoprotein B (gB, UL27). (B) One variant (causing a synonymous mutation at amino acid (AA) position 877 in gB) was detected in four v42 samples spanning two shedding episodes of this source partner. Similar fluctuations at this locus were also detected in three v43 samples, spanning three shedding episodes of this recipient partner. In both partners, early samples showed a dominant G nucleotide, which shifted to a dominant A nucleotide in later samples. In later v43 samples (see Table 2), this position harbored 100% penetrance of the "A" variant. (C) A second site of within-host diversity in gB was specific to the viral population of v43. This minor variant encoded a nonsynonymous mutation at AA position 175 in gB. This variant was present at a high frequency (65%) in the site-specific lesion sample on day 349, but it was detectable at a far lower frequency (4%) in a genital area swab collected on the same day. The frequency of this variant dropped to 7% in the day 352 lesion sample, and it was undetectable in that day's genital area swab.

variant detected at a ratio of 95% T to 5% G (a silent variant at AA77 in UL2). In the site-specific genital lesion from the same day (v43_d349_gen_les), the ratio of nucleotides is 40% T to 60% G at this location in the UL2 gene, echoing the frequencies of the UL27 (gB) variants in the same samples (Fig 5C). The parallel frequency of the within-host diversity in the UL2 variant and the UL27 variant suggests the potential co-segregation of viral genotypes. Two additional non-synonymous MVs were detected in two other sets of genital area swabs and lesion samples from participant 43 (at days 350 and 352). These MVs included a Val427Gly switch in VP 13/14 (encoded by UL47) at day 350, and an a Ala44Thr mutation in ICP34.5 (encoded by RL1) at day 352. Both of these variants showed small shifts in frequency between samples, but the minor variants remained ≤ 10% (see S2 Table for details). The ICP34.5 mutant was also detected in an earlier genital sample on day 349, but it was not detected in the same-day genital lesion sample. These data provide additional examples of differences in variant frequency between larger samplings of an entire anatomical area (i.e., genital swabs) versus site-specific lesion samples. The observed patterns of within-host variation demonstrate rapid changes in

genetic diversity within single individuals, and the potential for stochastic transmission of a subset of that diversity.

## High within-host HSV-1 diversity can be shared between transmission partners, and across anatomical niches

In this study, we were able to compare the presence or absence of within-host viral genetic diversity between partners and across anatomical niches. In transmission Pair 1, the source partner, participant 40, reported first symptoms of oral HSV-1 infection 14 years prior to transmission of HSV-1 to the recipient partner, participant 41. We sequenced three samples from participant 40 (one genital sample and two oral samples), and two samples from participant 41 (one genital and one oral sample). In the source v40 genomes, within-host diversity was only detectable in the first of two oral samples (v40_y14_oral1) with 27 MVs identified and all of those being below 4% frequency (Table 2 and Fig 6A; see S2 Table for full list of MVs). These MVs spanned positions across the entire genome, and included synonymous, non-synonymous, and intergenic variants (Fig 6A). Of the 27 variants, 11 were close enough in proximity to be connected on a single sequencing read to another nearby variant (Fig 6C). Within-host diversity was only detected in one of the recipient's v41 samples (v41_d354_oral), a partial genome from an oral swab with an average coverage depth of 28X and gaps in 3% of the genome. For this lower-coverage sample, we only considered within-host diversity $\geq 20\%$ frequency (10X minimum coverage depth; gaps excluded), which led to a total of 530 MVs detected (Fig 6B). Again, MVs were distributed across the genome, and 313 were connected on individual sequence reads (Fig 6D). Thus in Pair 1, we only detected within-host viral genetic diversity in the oral niche of either partner, and the presence of variants detected on the same individual sequencing read (Fig 6C and 6D) suggested a potential for co-segregating genotypes.

To explore potential co-segregation and transmission of MVs in Pair 1, we compared the position and identity of each MV to check for shared variants between both participants. In addition to looking at within-host MVs detected in both participants, we also considered whether a variant detected in one participant might be present in the other partner, but be below our minimum threshold for detection. This is especially relevant due to the lower coverage of the v41 oral sample (28X). This analysis indicated that all of the variants detected in source v40 genome (n = 27) were also detectable in recipient v41 genome, either above or below the adjusted threshold for MV detection (Fig 6E). Similarly, 411 (76%) of the variants detected in the recipient v41 genome were also detectable in the source v40 genome, above or below the 2% standard threshold (Fig 6F). While this indicates that many of the MVs were likely transmitted between partners, it cannot reveal the directionality of transmission. In the source v40's first oral genome (v40_y14_oral1) the 428X coverage depth was sufficient for robust MV detection, but the overall frequency of MVs was low. In the recipient's v41 genome, their oral sample (v41_d354_oral) had low and variable coverage depth which required a stringent filter that may have limited MV detection. These challenges mean that additional variants may simply be below the threshold for detection (i.e., in regions with <10X coverage depth). In contrast, the many variants that are detected and shared in both partners are notable and unlikely to occur in these samples by chance.

A similar phenomenon occurred in transmission Pair 4, between source participant 46 and recipient participant 47. However, in this case the samples illustrating high within-host genetic diversity came from different anatomical niches for each partner (Table 2, see S2 Table for full list of MVs). Participant 46, the source partner, had both oral and genital HSV-1 samples sufficient for sequencing. The six genital niche samples indicated very low within-host viral

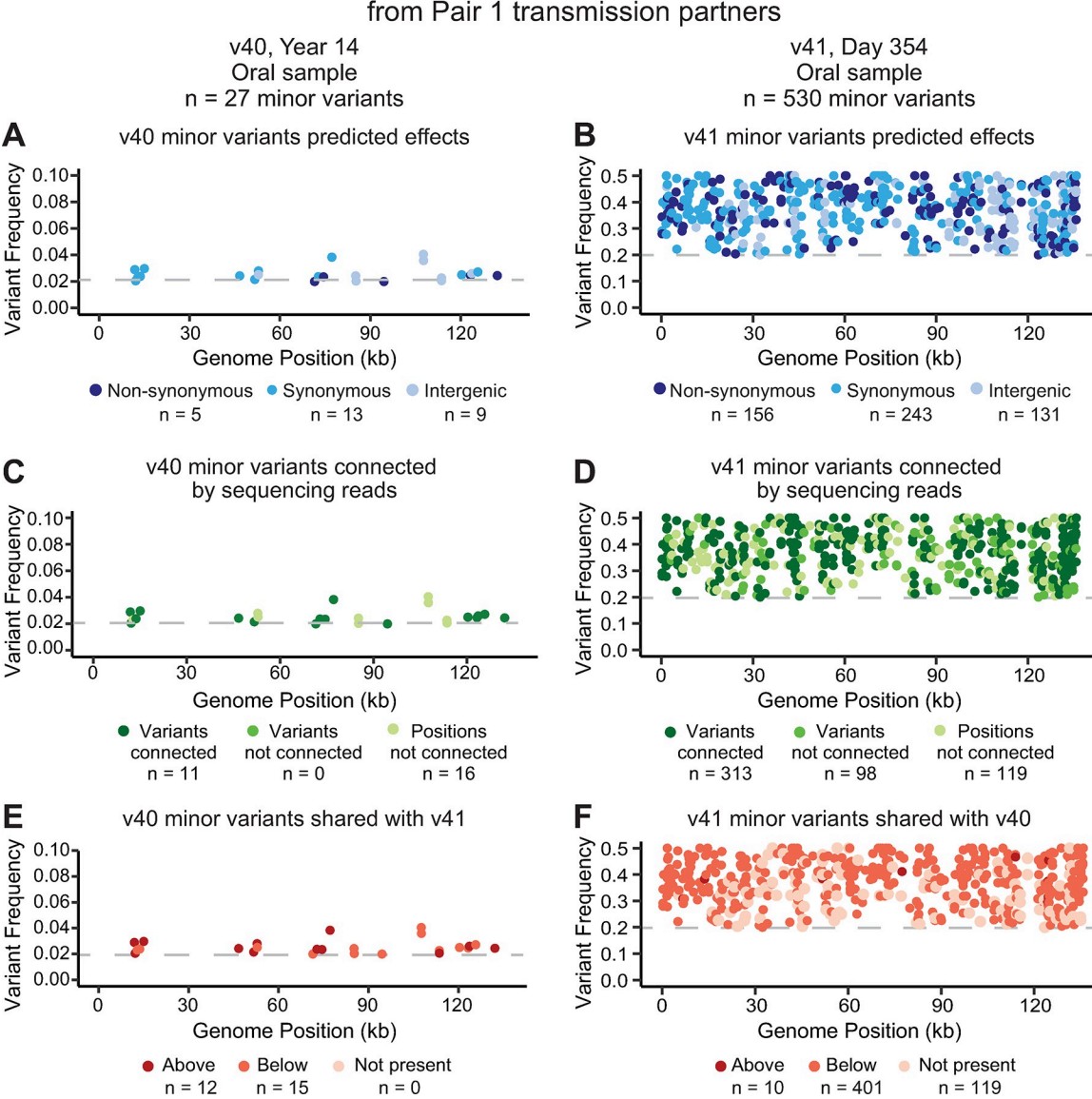

**Fig 6. Oral samples from Pair 1 indicate shared within-host genetic diversity.** Both the source and recipient partners in Pair 1 harbored high levels of within-host HSV-1 genetic diversity, or minor variants (MV), in at least one sample each. The source sample v40 had an average coverage depth of ~428X, allowing for MV analysis with a 2% threshold (v40_y14_oral1; 27 MVs detected). The recipient's sample v41 had an average coverage depth of 28X, and thus a stringent threshold that required ≥ 20% MV frequency was applied (v41_d354_oral, 530 MVs detected). **(A-B)** In both cases, MVs were randomly distributed across the HSV-1 genome, with a range of impacts from synonymous to nonsynonymous to intergenic. **(C-D)** In the v41 sample, a majority of variants were observed to be connected to nearby variants on the same sequencing read. **(E-F)** Most MVs were shared between partners; colors indicate if each MV was above threshold, below threshold, or not detected in the transmission partner. See S2 Table for position and frequency of all MV loci.

diversity, with only three samples harboring one MV each outside of repetitive areas (at a 2% detection threshold). Likewise, within-host diversity analysis using a stringent 20% threshold on the only oral sample available from this participant (v46_y16_oral; average coverage depth of 34X), detected 5 MVs in areas of the genome with ≥10X coverage depth (Fig 7A). The frequencies of these MVs ranged between 20–35% and were distributed across the genome (Fig 7A). None of these MVs were close enough in position to be observed on the same sequencing

## Comparison of viral within- & between-host genetic diversity from Pair 4 transmission partners

**v46, Year 16**
**Oral sample**
**n = 5 minor variants**

**v47, Day 61-79**
**Genital area samples pooled**
**n = 256 minor variants**

**Fig 7. Samples from Pair 4 indicate shared within-host genetic diversity across anatomical niches.** Both the source and recipient partners in Pair 4 harbored high within-host HSV-1 genetic diversity in at least one sample. In this comparison, both samples had an average coverage depth ~30X, so a stringent threshold requiring $\geq$ 20% MV frequency was applied when detecting MVs (v46_y16_oral had 34X coverage and 5 MV detected, while v47_d61-79_gen had 31X coverage and 256 MV detected). **(A-B)** MVs and their predicted effects were randomly distributed across the HSV-1 genome. **(C-D)** In the v47 sample, a majority of variants were connected to other nearby variants by the same sequencing read. **(E-F)** Most MVs were shared between transmission partners; colors indicate if each MV was above threshold, below threshold, or not detected in the transmission partner. See S2 Table for position and frequency of all MV loci.

read (Fig 7C). All samples available from the recipient partner (v47) were low in viral genome copy number, with less than 1000 genome copies in each sample (3 $\log_{10}$ copies per mL). Therefore, we pooled four genital swab samples spanning two shedding episodes within the same month, creating v47_d61-79_gen (Table 2). This combined sample yielded enough viral

genetic material for a genome with an average coverage depth of 31X and gaps in just 1.5% of the genome. Using the stringent 20% threshold for MV detection, we found 256 MVs distributed across the genome in areas of ≥10X coverage, with frequencies between 20–50% (Fig 7B). Among the closely positioned MVs, 155 were present on the same sequencing read as another variant (Fig 7D). Although we cannot discern which MVs originated from which of the four pooled samples for the recipient partner, the MVs co-occurring on individual sequencing reads would necessarily have been shed on the same day. Next, these samples were compared to determine if any MVs were shared across the transmission pair. All five of the MVs in the source partner's oral sample were shared with the recipient's pooled genital sample, either above or below threshold (Fig 7E). In the reverse comparison, 194 MVs (76%) in the recipient's pooled genital sample were shared with the source partner's oral sample (Fig 7F). While both samples in this comparison are low-coverage, the existence of this much shared MV diversity between partners (77–100% shared MV) strongly suggests that these variants are not random sequencing errors. Instead, these minor variants likely represent viral genetic diversity that is at the boundaries of current technology for DNA capture and sequence detection. The shared variants observed between the Pair 4 partners illustrate that high levels of within-host diversity can be transmitted as a population to new hosts.

In both the Pair 1 and Pair 4 examples above, only one sample from each participant harbored a high level of within-host diversity. This left open the question of how long MVs may persist during a given shedding episode. In Pair 5 the source partner, participant 48, harbored an infection with a high number of MVs. In these v48 samples, we were able to track the viral MV profile over the final four days of one particular shedding episode (Fig 8). Within-host diversity was apparent in the first two of four oral samples collected between day 100–104 (v48_d100_oral and v48_d101_oral). These samples had counts of 62 and 234 MVs, respectively, with coverage >350X (Fig 8A and Table 2). The MV frequencies ranged from 2–20% (Fig 8B and Table 2) and were distributed across the genome (Fig 8D and 8E). As done for similar analyses of Pair 1 and Pair 4 samples above, we examined whether or not any of these MVs co-occurred on the same sequencing read. We found that 100% of the MVs in sample v48_d100_oral and 92% of those in sample v48_d101_oral were connected with another MV on the same sequencing read. Many of the MVs present in the d100 sample were among those detected on d101 also (S2 Table). The final two days of sampling within this shedding episode revealed no within-host diversity (zero MVs) in the two subsequent oral samples (v48_d102_oral and v48_d103_oral). Interestingly, all four samples were collected during the peak of viral shedding for this episode (Fig 8C). This four-day stretch of sampling showed a reduction in the number of MVs as viral load was increasing, demonstrating that these phenomena are not directly coupled. These MVs were not detected in the viral population sequenced from either of two genital lesion samples available from the recipient partner, participant 49, both of which were collected several months later. These data illustrate that in some cases, within-host diversity may fluctuate in the observed viral population over the course of a multi-day shedding episode.

## Discussion

In this study, we provide a first look at the diversity and stability of HSV-1 genomes between adult sexual transmission partners. This analysis focused on the first year of genital HSV-1 infection of the newly-infected recipient partner. We anticipated that the viral population may be in flux in the first year of infection, due to transmission bottlenecks with subsequent population expansion and spread, or due to selective pressure(s) as the new host's immune response develops. Our data describe five transmission pairs, in which we compared viral genome

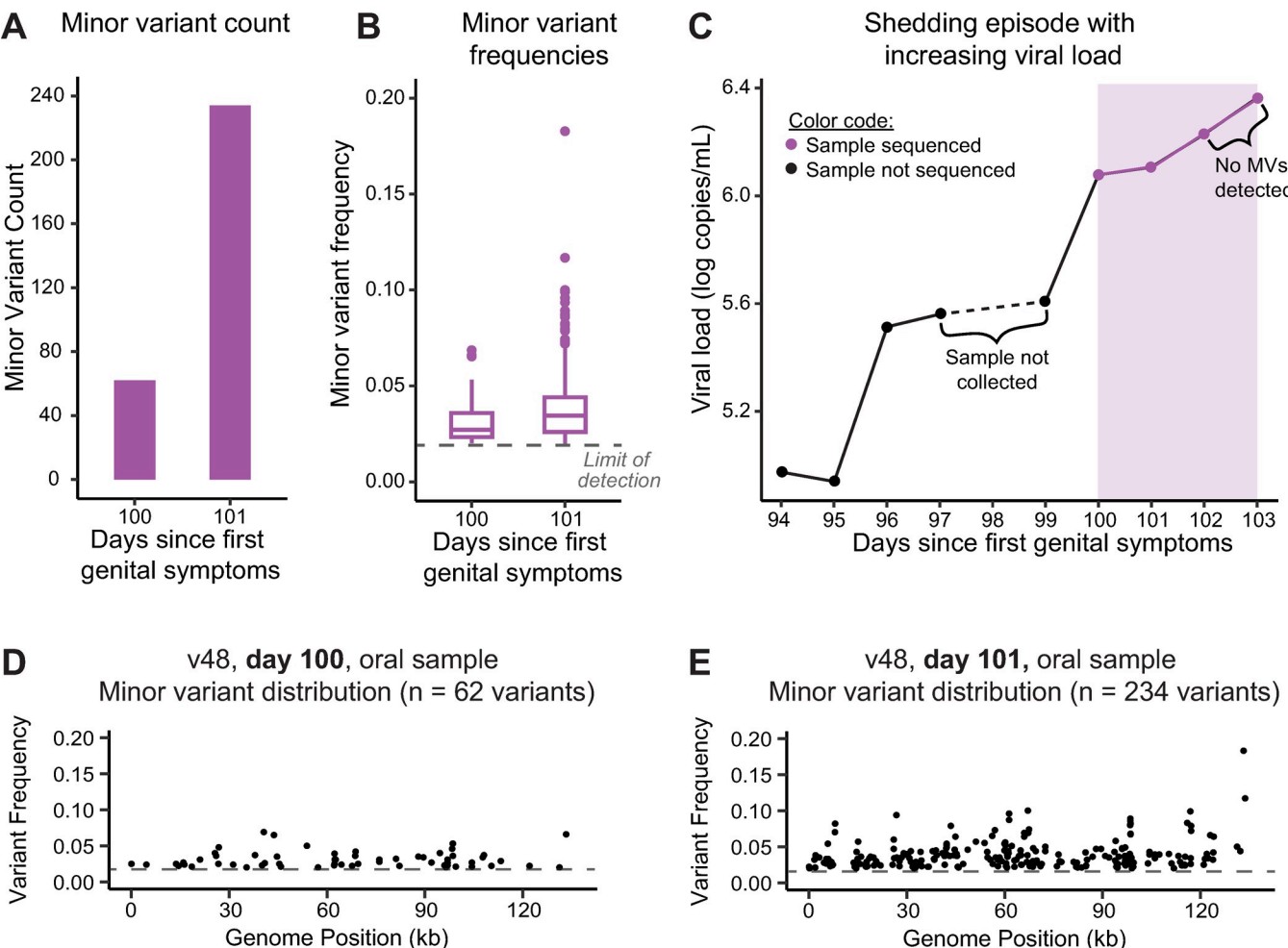

**Fig 8. Within-host diversity in source participant v48 varies over time within a single shedding episode.** Within-host diversity analysis of v48 oral samples on days 100–103 since first symptoms revealed minor variants (MV) only on days 100 and 101. **(A)** The overall number of minor variants increased from days 100 to 101 (v48_d100_oral had 350X coverage and 62 MV detected, while v48_d101_oral had 466X coverage and 234 MV detected). **(B)** The individual frequencies of these MVs remained between 2–20%. **(C)** These samples were collected at peak viral load (indicated by lavender shading), at the end of this shedding episode for participant 48. The following two oral samples (on days 102–103) had zero MVs detected. **(D-E)** The MVs in samples from day 100 and 101 were randomly distributed across the HSV-1 genome. Many of the same MV loci detected in the d100 sample were also detected on d101. See S2 Table for position and frequency of all MV loci.

sequences sampled from viral shedding in both the oral and genital niches over a one-year period. Each set of transmission-pair viral genomes occupied a unique branch in a network graph containing 65 other HSV-1 genomes. We found ≥ 98% nucleotide identity between viral genomes at the consensus-level within each transmission pair. We also noted single nucleotide differences (or SNVs), occurring at the consensus level in three out of five pairs (Pairs 1, 2, and 4). Several of these differences arose over distinct shedding episodes, while others fluctuated within a single episode, as detected in participant 43's genital samples (Pair 2). Across all participants and specimen types, we found within-host viral genetic diversity to be limited and often transient. The quantity of within-host diversity, or minor variants (MVs), was similar between transmission partners, although in several cases MVs were differentially

detected between site-specific lesion samples vs. those encompassing the entire genital area. These data reveal that consensus-level nucleotide changes can occur after transmission of genital HSV-1, and that minor variant frequencies can fluctuate over time and between shedding episodes. The ability for HSV-1 to undergo genetic drift and/or selection during transmission is likely driven by the extent of within-host diversity present during a particular shedding episode, as well as viral genetic fitness with respect to each host.

When HSV-1 infections have been sampled from unrelated individuals, we and others have found these to be genetically distinct [6,7,9,10,23,30]. However, in prior analyses of familial transmission pairs, an extremely high level of conservation was observed between parent and child transmission partners [7,30]. The high percent nucleotide identity between transmission pair samples in this study (with >98% conservation within each pair), regardless of sampling time, suggests that non-familial transmission events are almost as well-conserved as familial-transmission events. Among the sequenced genomes in this study, three out of five transmission pairs did have consensus-level differences or SNVs (outside of repetitive regions) (Pairs 1, 2, and 4) (Fig 3). The range of SNVs between partners (1–300 nucleotide sites) caused only a minimal shift in the co-location of partner samples in the global network graph analysis (Fig 2). This fidelity at the consensus genome level is similar to that observed in the only two prior reports of longitudinal HSV-1 [5] and HSV-2 [28] clinical genomic studies. However, a closer examination of these nearly conserved consensus genomes (Fig 3) reveals that nucleotide changes can occur transiently over brief time periods (v43 genital lesion vs. genital area specimens; Fig 5C) and can differ between oral and genital niches (Pair 1 and 4; Fig 3). We hypothesize that transient viral genetic diversity may be localized to specific areas within each shedding episode, perhaps influenced by localized CD4+ T cell populations [34–36]. The expanding viral population of each shedding episode may also change in relation to the amount of within-host diversity initially seeded in the ganglia, and subsequent variability generated during reactivation [12,37,38]. It would be ideal to study additional transmission pairs to explore how these factors may affect consensus genome diversity. We also noted that transmission pairs with no single-nucleotide differences between partner genomes can still provide insights into the viability of rare consensus-level variants. The alternative gH tail encoded in Pair 3 bolsters previous studies suggesting that insertion/deletion variants may broaden the impact of HSV-1 genomic variability [10,39,40]. These findings warrant future studies to explore the breadth and accumulation of natural HSV-1 diversity in relation to an individual host's immune response and shedding patterns.

We hypothesize that the genetic diversity of the recipient partner's viral population is dependent on multiple factors including the level of within-host viral diversity in their source partner, the quantity of virus shedding at the mucosa during transmission, and the extent of a population bottleneck and subsequent expansion in the new host. Within-host diversity of this type can also be described as standing variation in the viral population. The large percentage of shared MVs between transmission partners in pairs 1, 2, and 4 illustrates the influence of standing variation in the source partner's viral population (Figs 5–7 and S2 Table). These data are consistent with the shared within-host diversity observed in a recent case study of viremic HSV-1 transmission from mother-to-neonate [7]. Moreover, we noted that the source partners in Pairs 2 and 4 had a high rate of shedding (i.e. greater than 25%) in the second session (Table 1). This suggests that these source partners may be an exception to the typical patterns of genital HSV-1 disease, since most individuals experience reduced shedding by the end of the first year of infection [31,41,42]. We cannot rule out the contribution of ongoing transmission of MV diversity from the same source population, although the establishment of host immune responses may limit this. Lastly, the ability of viral genotypes to be transmitted is also known to be impacted by population bottlenecks and subsequent expansion. The mother-

neonate transmission event suggested a wide or loose bottleneck, perhaps due to maternal viremia in this unusual case of dual fatality [7]. Despite the differences in transmission route and quantity of shed virus, the current data from adult sexual partners agree with a wide population bottleneck. Prior models of human cytomegalovirus (HCMV) transmission *in utero* have suggested a population bottleneck exists during transmission, followed by population expansion in neonatal niches [43]. HCMV infections tend towards greater genetic diversity than HSV-1, as the virus replicates within more cell types, is more widely disseminated, and undergoes frequent recombination during superinfection with multiple genotypes [44,45]. For HSV-1, the complexity of interacting factors will likely become clearer as additional studies seek to combine clinical characteristics of infection with viral genome sequencing analysis.

The deep-sequencing of clinical isolates collected over time in our study also provided the opportunity to explore the potential for transient flux in levels of within-host diversity. In one example, we detected minor variants undergoing rapid genetic change within a site-specific genital lesion from participant 43, over just three days (Fig 5C). A similar rapid change in MV prevalence over just three days of viral shedding was noted in a prior pilot study of a participant who was five years into her genital HSV-1 recurrences [5]. The gB variant in Fig 5C was robustly detected in the site-specific v43 genital lesion, but barely above threshold in the genital area samples collected at the same time. This variant was not detected in any of the source partner's samples. Potential sources of the rapid change in nucleotide identity (encoding a valine at amino acid position 175 in gB) might include a pre-existing viral genome variant that reactivated from latency, or positive selection due to site-specific T-cell surveillance [46]. Valine 175 in gB has never been observed at the consensus-level, in any prior published HSV-1 protein sequences. Homology-based structure modeling via PyMol does not predict that any significant structural changes would occur with a phenylalanine to valine change at this site [47]. It is interesting to note that the total amount of virus shed in the simultaneous swab of the entire genital area (which includes the lesion site) appeared to dilute out the MV that was detected in the site-specific lesion swab (Fig 5C). This suggests that when lesions are present, it will be important to continue sampling both the lesion site and the surrounding tissues to better understand how local shedding sites differ in their spatial and genetic complexity.

While the prior example demonstrated how spatial aspects of sample collection can influence the detection of minor variants in the viral population, the timing of sample collection also plays a role. In the case of the v48 oral samples highlighted in Fig 8, only two of the four peak-shedding days harbored minor variant diversity. If we had only sequenced the final sample from this episode, we would have missed the indication of high within-host genetic diversity at earlier days. Together, these data illustrate how infections can harbor localized, transient genetic variation during a shedding episode. It also demonstrates how limited sampling, and/or broad skin-swab approaches that survey a large area at once, may obscure underlying patterns of spatially-localized viral genetic diversity. Prior data had likewise established that viral shedding varies across the spatial area of the oral and genital niches, and these minor variant findings are in line with those shedding data [5,18,19]. These spatial and temporal aspects of sample collection may influence the detection of mixed or dual-infections. Prior studies using less sequence coverage depth and/or culturing viruses before sequencing, may have led to a reduced detection of within-host viral genetic diversity [5,7,16,23,28,30]. The pairing of direct-from-participant (uncultured) specimens collected from localized anatomical areas with deep sequencing, will be critical to decipher viral differences between lesion and non-lesion sites, and other potential correlates of specific viral genotypes [12].

Among the samples analyzed here, levels of within-host diversity were either very low (i.e., less than 10 MVs) or substantially higher (i.e., greater than 50 MVs) and distributed across the viral genome (Table 2). The temporal samples and availability of partner samples allowed us to

examine these higher-diversity samples in a broader context. In two out of three pairs with one partner showing high levels of minor variants (Pairs 1 and 4), we found that the vast majority of these MV loci were detected in both partners (Figs 6E, 6F, 7E and 7F). In these samples, MVs that were within 300 base pairs of one another were often connected on the same sequence read (Figs 6C, 6D, 7C and 7D). These data suggest a loose population bottleneck which allows the transmission of viral population diversity between partners. The proofreading ability of the HSV-1 DNA polymerase, and the lack of any apparent polymerase mutations in these samples make it unlikely for the observed number of MVs to reflect a sudden explosion of within-host diversity. Moreover, the distribution of numerous MVs across the viral genome, connected across a majority of sites, are suggestive of a mixed or dual HSV-1 infection with two (or more) distinct viral genomes. This hypothesis was also suggested in a recent study by Lassalle and colleagues, to explain the high genetic diversity detected within a genital swab of a 65 year old individual in their study [9]. The authors proposed that high genetic diversity could reflect multiple lineages of genomes undergoing recombination [9]. Such cases may be difficult to detect if recombination has shifted MV frequencies and altered the composition of the viral genome population [9,12].

The samples in this study reveal apparent transmission across oral and genital niches, exemplifying the recent trend toward HSV-1 causing new primary genital infections [1]. Interestingly, the high shared diversity exemplified in Pair 4, with transmission of linked minor variants, include an oral sample (v46) from the source partner and a genital sample (v47) from the recipient (for whom no oral shedding occurred during this study) (see Table 1). The changing epidemiology of HSV-1 infections may be increasing the rate of oral-genital mixing of strains and creating more opportunities for dual-infection. The oral infection of source partner 46, from a long-running infection (16 years prior), suggests the potential for a newly acquired genital infection genotype to have intersected with a pre-existing oral HSV-1 genotype (Fig 7). The opportunity for niche transfer and viral recombination may also be enhanced by repeated encounters with the same source partner. These observations bring into question the frequency with which dual-infections occur naturally in the population, and how often distinct viral strains are concurrently reactivated within a host [11,12]. In future studies, it would be useful to compare the age of infection acquisition to viral genetic diversity, to determine if dual-infections can accrue over the individual's lifetime or are only introduced during transmission to immunologically naïve hosts.

The world wide spread and prevalence of HSV-1 infections provides ample opportunity for dual-infections to arise, even if they are rare and/or transient events. Observations from prior studies suggest that HSV-1 genomes undergo frequent, contemporary recombination [24,48–51]. The potential for viral recombination when divergent viral genomes co-occur would provide a molecular mechanism by which a virus with a low polymerase error-rate could nonetheless accumulate and perpetuate extensive genetic diversity between hosts. Several of the highest-diversity samples in this study were collected from oral shedding in pairs where the source partner had a long-running prior infection in the oral niche (Pairs 1 and 4; Figs 6 and 7), raising the possibility that the oral niche, the long prior duration of those infections, and/or the coincidence of oral and genital infections in a single host could also be a factor in these observations. Additional sampling of transmission partners, particularly those where one partner is several years into their infection or presents with a pre-existing oral HSV-1 infection, will provide more insight into the frequency and transmission of dual or mixed infections. Moreover, as long-read sequencing techniques continue to be refined for low-input clinical samples, these methods can be applied to achieve better resolution of distinct genotypes.

As we have shown here, clinical HSV-1 can be deeply sequenced to obtain data on the within-host diversity and viral genetic variants that arise within each participant. The transient

genetic diversity detected from HSV-1 shedding episodes in this study suggest that these DNA viruses have the potential to undergo rapid adaptation. Future studies should examine the total diversity in viral shedding samples, as well as the diversity in localized areas (i.e., lesions). This may illuminate how these viruses escape from local immune pressure but maintain overall conservation between transmission partners. This viral diversity can also be explored for its potential significance in host-specific adaptation, including correlations with T-cell and/or B-cell responses, and functional studies to elucidate the functional impacts of *in vivo* HSV-1 genotype shifts. The within-host genetic diversity of HSV-1 should also be carefully examined for patterns that reflect multiple genotypes or recombination, versus positive selection or *de novo* variation.

## Methods

### Ethics statement

The study was approved by the University of Washington Human Subjects Division (IRB no. STUDY00001465). Participants provided written informed consent prior to study procedures.

### Participant and sample collection

Of the 88 participants enrolled in a large natural history study of first-episode or primary genital HSV-1 infection at the UW Virology Research Clinic (VRC) in Seattle, ten participants enrolled as either a "source" (i.e., transmitting) or "recipient" (i.e., newly-infected) partner of a transmission pair, and had HSV-1 detected from genital or oral swabs (Table 1) [31]. All participants were HIV negative and HSV-2 negative. The infection status of each participant was determined by screening for the presence (non-primary infection) or absence (primary infection) of HSV-specific IgG antibody, followed by HSV-specific Western blot to confirm development of the antibody response over time (for primary infections) [52]. The infection status at screening was designated "unable to determine (UTD)" if sera were collected more than four weeks after the participants' reported first episode and the sera tested positive for HSV-specific antibodies (Table 1). For those enrolling in the year-long study, swab and culture specimens were collected at the time of enrollment, with further self-collection (i.e. daily at-home collection) of oral and genital swabs in two 30-day sessions, occurring at 2 months and 11 months after their first-episode lesions [31]. The transmitting or "source" partners were identified and referred to the research clinic by persons with primary genital HSV-1 (i.e. the newly-infected "recipients"). Source partners were HSV-1 seropositive and enrolled in a study to collect a single 30-day session of oral and genital swabs (see below for details). None of the participants enrolled in this study were using antivirals during the 30-day collection periods, however antivirals were available for use as needed for symptom control during intervening months [31].

During each 30-day series of at-home swab collection, participants were instructed to rub a sterile swab over the entire genital region (i.e. vaginal or penile area, followed by external anal area; this encompasses the entire anogenital area), as in prior studies of genital HSV shedding [16,53]. For surveillance of oral shedding, participants were instructed to rub a sterile swab across the entire oral area (i.e. external surface of lips, followed by internal mouth areas of the tongue and cheek; this encompasses the entire orolabial area) [54]. Participants who experienced symptomatic genital recurrences were seen at the clinic for collection of site-specific genital lesion swabs, with documentation of lesion location. No oral lesions were observed in the course of the study. Each swab was placed into 1X PCR buffer solution as previously described, for subsequent quantitation of HSV genomes by qPCR for the gB gene (UL27) [15,18,55]. The percent of days positive by qPCR for HSV-1 was reported for participants for each 30-day session of daily swabbing as previously established [16,17,19]. Data were assessed

separately for oral vs. genital niche shedding, and for the first 30-day session vs. the late-year session (Table 1 and Fig 1B) [31]. Samples with HSV-1 DNA detected were selected for sequencing based on their temporal distribution over the first year of infection. Samples with less than 10,000 virus genome copies (4 $\log_{10}$ copies per mL) have generally yielded insufficient viral DNA using current sequencing methods, thus we prioritized samples with higher copy numbers. We also limited sample selection to those that would optimize sequencing cost and effort while spanning multiple shedding episodes (i.e., comparison from the beginning vs. the end of the first year of infection). In the case of participant 47, no samples were above 10,000 genome copies, so we pooled four samples to create a baseline consensus viral genome for this individual (Table 2). For several low-coverage samples, we re-sequenced the source libraries and concatenated reads from two sequencing runs to improve the coverage depth for genome assembly and minor variant analysis (Table 2). Each participant was given an arbitrary participant ID and viral genome ID. Even-numbered participant IDs indicate transmission source partners, and odd-numbered participant IDs denote recipient partners.

## DNA extraction, HSV qPCR, sample library prep, and Illumina sequencing

Swabs selected based on the criteria above were processed for target enrichment and sequencing according to our published protocol [56]. Briefly, samples were processed through a phenol:chloroform separation followed by ethanol precipitation for DNA extraction, with subsequent quantification of total DNA by Qubit and HSV genome copies by qPCR for the gB gene (UL27) [55]. Samples were then sheared into approximately 500–1,000 base pair fragments and processed for library preparation with the KAPA Biosystems HyperPrep Library Kit according to the manufacturer's protocol (with 14 cycles of amplification). Custom HSV-specific oligonucleotide probes (myBaits) from Arbor Biosciences were used with the Arbor Biosciences myBaits Target Capture Kit to enrich for HSV-1 DNA material according to the manufacturer's protocol, followed by a second round of amplification (14 cycles) using the KAPA HiFi HotStart Library Amplification Kit [56]. A final round of quantification by Qubit and gB-specific qPCR was performed and used to adjust each sample to the appropriate concentration for sequencing with an Illumina MiSeq, using version 3 chemistry and 300 × 300-bp paired-end reads.

## Virus genome *de novo* assembly

Prior to genome assembly, FASTQ data were filtered to positively-select for any HSV-specific reads. To do this, sequence reads were positively selected (PS) using a BLAST search of an HSV database containing all HSV genes and genomes in GenBank, and reads matching HSV with an e-value $< 10^{-2}$ were retained. These PS reads were then processed through the first step of the Viral Genome Assembly (VirGA) pipeline for quality control [57]. Briefly, the PS reads were screened using Trimmomatic [58] to remove artifacts and any Illumina adapter sequences, and trimmed to remove low quality bases (minimum Phred score 30, over a 15 bp window size). This process also removes any short read fragments (minimum size 30 bp) and any unpaired reads. Paired-end reads were then used for *de novo* consensus genome assembly using MetaSpades v.3.14.0 (parameters: *spades.py -k 21*, *33*, *55*, *77—meta -1 $R1–2 $R2 -o metaspades_output*) [59]. The metaspades scaffolds were then used for steps three and four of the VirGA pipeline, which included assembling a full draft genome and filling of gaps using GapFiller [60]. Sequenced reads were aligned back to the consensus draft genome via Bowtie2 for downstream analysis such as minor variant detection (see details below). Gene feature annotations were identified by homology comparison to HSV-1 strain 17 as the reference sequence (GenBank Accession JN555585) [57,61].

## Consensus genome comparison and phylogenetic analysis

Trimmed consensus genomes were used for all comparisons, meaning that the terminal copies of the repeat long/short (TRL/TRS) regions of the HSV-1 genome were not included. The trimmed consensus of all 33 viral genomes were aligned using MAFFT v7.394 with default parameters to create pairwise global nucleotide alignments [62]. Amino acid comparisons were performed by generating alignments via ClustalW2 [63]. Network graphs of multiple sequence alignments were constructed via SplitsTree v4 using the uncorrected P-distance and all gap regions removed [64]. S1 Table contains a list of strain names, GenBank accession numbers, and references for the previously published 65 HSV-1 genomes in the network graph analysis. Nucleotide and amino acid sequences were compared and visualized using Geneious software v11 [65]. Custom python scripts were used to extract single-nucleotide difference or variant (SNV) counts from each pairwise comparison of consensus-level genomes, and these were subsequently manually curated to exclude SNVs at repetitive elements or insertions and deletions (indels).

## Minor variant detection

We examined sequence reads for within-host nucleotide variability by first using Picard RemoveDups to remove PCR duplicates from the Bowtie2 aligned reads to the consensus genome build, and then applying VarScan v2 with SnpSift/Eff [61,66–69]. MV parameters for samples with an average coverage depth $\geq$ 100X were as follows: minimum coverage depth: 100X, minimum variant reads: 2, minimum variant frequency: 0.02 (2%). For five of the 33 samples, the viral genome coverage depth was too low for our initial parameters for detecting MVs. Thus, we implemented a more stringent detection threshold for the five samples with an average coverage depth $<$ 100X, with parameters as follows: minimum coverage depth: 10X, minimum variant reads: 2, minimum variant frequency: 0.2 (20%). A custom python script was then applied to filter all MV calls for strand bias (threshold: no more than 90:10). MVs were also manually curated to minimize potentially false-positive calls due to misaligned sequence reads that can occur at the edges of homopolymers and other repetitive elements. Shared MVs between transmission partners were identified by aligning sequencing reads from a source partner's sample to their recipient partner's consensus genome to create an alignment for cross comparison (and vice versa). Shared MVs that were above detection thresholds in one partner, but designated as "below threshold" in the second partner (Figs 6 and 7) were identified by the presence of at least one read with the shared variant in the second partner's sample. Minor variants detected in all samples are listed along with their quantitative metrics in S2 Table.

## Statistics

The percent of days with HSV detected was determined for each participant with at least one day of self-collected swabs by calculating the proportion of days with HSV detected out of all days with swabs collected, as previously described [16,17,19]. Minor variant analysis was performed using VarScan 2, an algorithm incorporating user parameters (as described above) and a default setting of p-value $\leq$ 0.05 to assess the statistical significance of each variant by Fisher's exact test [69].

## Supporting information

**S1 Table. List of previously published HSV-1 genomes used for phylogenetic analyses.** This file contains a list of strain names, GenBank accession numbers, and references for the

previously published 65 HSV-1 genomes used in the network graph analysis in Fig 2. (PDF)

**S2 Table. List of minor variants (MVs) detected in viral genomes in this study.** MVs detected in all viral genomes (as listed in Table 2) are listed in this table, along with quantitative statistics such as the number of sequence reads supporting the major vs. the minor variant alleles, the precise genome position of each variant, and its annotation (e.g. synonymous, non-synonymous, or intergenic). This file provides the supporting data for Table 2 and Figs 5–8. (XLSX)

## Acknowledgments

We thank Daniel Renner for bioinformatics support, Ellie Kim for PyMol analysis of the glyco-protein B variants, members of the Heldwein lab for interesting discussion of glycoprotein B structure, and David Koelle and Lichen Jing for thoughtful discussions of HSV-1 immunology.

## Author Contributions

**Conceptualization:** Molly M. Rathbun, Christine Johnston, Moriah L. Szpara.

**Data curation:** Molly M. Rathbun.

**Formal analysis:** Molly M. Rathbun.

**Funding acquisition:** Anna Wald, Christine Johnston, Moriah L. Szpara.

**Investigation:** Molly M. Rathbun, Mackenzie M. Shipley, Christopher D. Bowen, Stacy Selke.

**Methodology:** Molly M. Rathbun, Mackenzie M. Shipley, Christopher D. Bowen, Christine Johnston, Moriah L. Szpara.

**Project administration:** Moriah L. Szpara.

**Resources:** Molly M. Rathbun, Stacy Selke, Anna Wald, Christine Johnston, Moriah L. Szpara.

**Software:** Molly M. Rathbun.

**Supervision:** Christine Johnston, Moriah L. Szpara.

**Validation:** Molly M. Rathbun, Mackenzie M. Shipley.

**Visualization:** Molly M. Rathbun, Moriah L. Szpara.

**Writing – original draft:** Molly M. Rathbun.

**Writing – review & editing:** Molly M. Rathbun, Mackenzie M. Shipley, Christopher D. Bowen, Stacy Selke, Anna Wald, Christine Johnston, Moriah L. Szpara.

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
