## [Decision Letter · Decision Letter 0]

15 Dec 2021

Dear Dr. Szpara,

Thank you very much for submitting your manuscript "Comparison of herpes simplex virus 1 genomic diversity between adult sexual transmission partners with genital infection" for consideration at PLOS Pathogens. As with all papers reviewed by the journal, your manuscript was reviewed by members of the editorial board and by two independent reviewers who are experts in the field. Both reviewers found the subject of the study to be significant. Whereas Reviewer 2 was very enthusiastic, Reviewer 1 had major concerns about data analysis and interpretation. In light of the reviews (below this email), we would like to invite the resubmission of a significantly-revised version that carefully addresses all of the reviewers' criticisms. In addition, to engage a broader PLOS Pathogens readership, please, restate your major conclusions throughout the manuscript avoiding jargon.

We cannot make any decision about publication until we have seen the revised manuscript and your response to the reviewers' comments. Your revised manuscript is also likely to be sent to reviewers for further evaluation.

Sincerely,

Ekaterina E. Heldwein

Guest Editor

PLOS Pathogens

Erik Flemington

Section Editor

PLOS Pathogens

Kasturi Haldar

Editor-in-Chief

PLOS Pathogens

orcid.org/0000-0001-5065-158X

Michael Malim

Editor-in-Chief

PLOS Pathogens

orcid.org/0000-0002-7699-2064

Reviewer's Responses to Questions

**Part I - Summary**

Reviewer #1: Both herpes simplex virus 1 and 2 (HSV-1 and HSV-2) can be responsible for genital herpes lesions. HSV-1 classically oral infections and HSv-2 genital infections, however HSV-1 epidemiology is evolving in favour of an increasing number of primary infections occur in the (ana-)genital anatomical niche. Viruses could adapt to their host over time thanks to genetic adaptation. As few data are known for HSV, especially for HSV-1 in the context of genital herpes, all studies concerning this subject are quite interesting. Then, the authors, Rathbun at al., try to study viral genetic diversity in the context of HSV-1 genital herpes. Indeed, they used deep-sequencing analysis to characterize the consensus HSV-1 sequences and to detect minor variants (MVs) within paired sexual partner samples.

Reviewer #2: Rathbun et al describe an important and elegant study of genetic variation in herpesvirus genomes among recently infected sexual partners. Both the study and the manuscript are highly accomplished and the authors are to be congratulated.

**Part II – Major Issues: Key Experiments Required for Acceptance**

Reviewer #1: I have read through the manuscript and I have major comments.

- To my opinion, the most bothering fault of this study is the analysis of sequencing data. Indeed how is it possible to take into account sequencing data whereas average coverage is very low (see Table 2: e.g. Sample v47_d61-79_gen with 31x of average coverage associated with low viral load of 2.6 cp/mL of virus as data was obtained form 15000 reads with pooled from two replicates or Sample v41_d354_oral with 28x of average coverage associated with low viral load of 2.2 cp/mL of virus as data was obtained form 4100 reads). The use of threshold of 20% supposedly more stringent is not appropriated. To my opinion, to be relevant, only samples with coverage depth <100x should not be considered. Most of the time, low average coverage is associated few reads available and high MV detection: e.g. Sample v47_d61-79_gen with 31x of average coverage associated with 225 MVs or Sample v41_d354_oral with 28x of average coverage associated with 144 MVs.

Can we trust that?

Moreover, the authors do not discuss quality criteria for sequence trimming in the MM section. May the authors cite publications where these criteria have been used for NGS analysis for HSV genomic diversity estimation?

- According to the authors, detection of MVs within the viral population revealed both shared and unique patterns of genetic diversity between partners, and between anatomical niches. But how to make the clear distinction anal and genital niche with self-sampling samples.

- Additionally, the authors discuss that genetic drift was detected from spatiotemporally separated samples in as little as three days. This period of time of 3 days appear to be unrealistic. Indeed, even under antiviral selection pressure for instance the evolution pf HSV-1 do not reach such a level of MV within viral genes implicated in antiviral resistance.

- It appears there is some discordance between text/tables/figure data makes difficult the interpretation of the study. Either I do not understand or this formulation is not clear enough for well-understanding! I think the authors simplify the tables and the figures to get straight to the point.

Reviewer #2: (No Response)

**Part III – Minor Issues: Editorial and Data Presentation Modifications**

Reviewer #1: NONE

Reviewer #2: There are a number of places where it appears the text is referring to the wrong figure number or sub-number eg 6b rather than 6 C on page 13 line 292. It would be useful for the authors to systematically review all such references for accuracy.

The discussion should include comments on the limitations of the methodology in terms of variation in the depth of sequencing coverage between samples and the implications of this on the ability to detect MV.

PLOS authors have the option to publish the peer review history of their article (what does this mean?). If published, this will include your full peer review and any attached files.

Reviewer #1: No

Reviewer #2: No
---

## [Editor Report · Decision Letter 1]

11 Mar 2022

Dear Dr. Szpara,

We are pleased to inform you that your manuscript 'Comparison of herpes simplex virus 1 genomic diversity between adult sexual transmission partners with genital infection' has been provisionally accepted for publication in PLOS Pathogens.

Best regards,

Ekaterina E. Heldwein

Guest Editor

PLOS Pathogens

Erik Flemington

Section Editor

PLOS Pathogens

Kasturi Haldar

Editor-in-Chief

PLOS Pathogens

orcid.org/0000-0001-5065-158X

Michael Malim

Editor-in-Chief

PLOS Pathogens

orcid.org/0000-0002-7699-2064
---

## [Editor Report · Acceptance letter]

27 Apr 2022

Dear Dr. Szpara,

We are delighted to inform you that your manuscript, "Comparison of herpes simplex virus 1 genomic diversity between adult sexual transmission partners with genital infection," has been formally accepted for publication in PLOS Pathogens.

Best regards,

Kasturi Haldar

Editor-in-Chief

PLOS Pathogens

orcid.org/0000-0001-5065-158X

Michael Malim

Editor-in-Chief

PLOS Pathogens

orcid.org/0000-0002-7699-2064